# Long-Term Monitoring of Cork and Holm Oak Stands Productivity in Portugal with Landsat Imagery

**Valentine Aubard \***, **Joana Amaral Paulo** and **João M. N. Silva**

Forest Research Centre, School of Agriculture, University of Lisbon, Tapada da Ajuda, 1349-017 Lisbon, Portugal; joanaap@isa.ulisboa.pt (J.A.P.); joaosilva@isa.ulisboa.pt (J.M.N.S.)
**\*** Correspondence: vaubard@isa.ulisboa.pt; Tel.: +351-213653387

**Abstract:** Oak stands are declining in many regions of southern Europe. The goal of this paper is to assess this process and develop an effective monitoring tool for research and management. Long-term trends of the Normalized Difference Vegetation Index (NDVI) were derived and mapped at 30-m spatial resolution for all areas with a stable land cover of cork oak (*Quercus suber* L.) and holm oak (*Quercus ilex* L.) forests and agroforestry systems in mainland Portugal. NDVI, a good proxy for forest health and productivity monitoring, was obtained for the 1984–2017 period using Landsat-5 TM and Landsat-7 ETM+ imagery. TM values were adjusted to those of ETM+, after a comparison of site-specific and literature linear equations. The spatiotemporal trend analysis was performed using only July and August NDVI values, in order to minimize the spectral contribution of understory vegetation and its phenological variability, and thus, focus on the tree layer. Signs and significance of trends were obtained for six representative oak stands and the whole country with the Mann Kendall and Contextual Mann-Kendall test, respectively, and their slope was assessed with the Theil-Sen estimator. Long-term forest inventories of six study sites and NDVI time series derived from the Moderate Resolution Imaging Spectroradiometer (MODIS) allowed validating the methodology and results with independent data. NDVI has a good relationship with cork production at the forest stand level. Pettitt tests reveal significant change-points within the trends in the period 1996–2005, when changes in drought patterns occurred. Twelve percent of the area of oak stands in Portugal presents significant decreasing trends, most of them located in mountainous regions with shallow soils. Cork oak agroforestry is the most declining oak forest type, compared to cork oak and holm oak forests. The Google Earth Engine platform proved to be a powerful tool to deal with long-term time series and for the monitoring of forests health and productivity.

**Keywords:** forest monitoring; *Quercus suber* L.; *Quercus ilex* L.; *montado*; agroforestry; time series; Normalized Difference Vegetation Index; Contextual Mann-Kendall; Google Earth Engine

## 1. Introduction

Cork oak (*Quercus suber* L.) and holm oak (*Quercus ilex* L.) cover one-third of the woodland areas in continental Portugal. They have been exploited for centuries [1] mostly as an artificial extensive silvopastoral system called *montado* [2]. Those two species have an important value for Portugal's economy, society, and environment, providing various forest ecosystem services as landscape and tourism hotspots, reducing fire risk and soil erosion, and increasing carbon sequestration and key-habitats for rare and endemic species [3–5]. Under the typical silvopastoral management system, those trees are good shelters for animals, either in winter against frost, or in summer against heat and sun exposure [3,6]. Holm oaks produce characteristic acorns that are used to feed livestock, in particular black pigs which are associated with these stands and produce premium meat, and culinary purposes. The wood is known to be extremely calorific and is traditionally used as firewood,

and, before the development of fossil fuel, for charcoal production [3,7]. Cork oaks are also exploited for their acorns, but mainly for their cork, which is one of the most valuable products of non-wood forest exploitation worldwide, having remarkable mechanic properties such as low permeability to liquids and gases, resistance to rot, and high elasticity [8]. It is used mainly for bottle stoppers, but also isolation material, clothing, and other products. Portugal is the main producer of cork in the world, owning 50% of the global market, and the sector contributes to 3% of the gross domestic product of the country [9], providing many jobs, especially in rural areas.

An important decline of oaks has been noticed in the last decades in Portugal due to various possible factors such as land use changes, especially livestock grazing [10], difficult access to groundwater in summer in hilltops and shallow soils [11], climate change [12], and new pests and diseases [13,14]. This decline has been observed in several ecosystem features such as the number of trees per hectare [15,16], tree regeneration, tree canopy cover [17], or cork growth [18]. To the best of our knowledge, the present study is the first to quantify and spatialize the current decline of oak woodlands at the national scale.

The Normalized Difference Vegetation Index (NDVI) has been used for monitoring purposes [19] and is considered an adequate proxy of the global health of holm oak and cork oak stands and correspondingly of cork and acorns productivity. A strong correlation was found between wood, leaves and seed production of North American oak species and NDVI derived from the National Oceanic and Atmospheric Administration Advanced Very High Resolution Radiometer satellite (NOAA/AVHRR, 1-km of spatial resolution) [20]. NDVI from the Moderate Resolution Imaging Spectroradiometer (MODIS, 250-m resolution) was also proved to be an efficient indicator of forest biomass growth and dieback in Mediterranean holm oak forests [21]. Few studies used NDVI or other vegetation indices to estimate land cover changes and vegetation trends in the Iberian Peninsula [22,23], pre- and post-fire vegetation dynamics [24], drought response of Mediterranean evergreen oaks [25], the relationship between their phenology and precipitation [26]. Also, the Enhanced Vegetation Index (EVI) derived from Landsat imagery was used to detect productivity trends of cork and holm oaks in Serra de Grândola, Portugal [27]. This study used a 13-year time series for a 20 km$^2$ area and a Mann-Kendall test to determine the significance of the trends. In our study, the analysis was extended to all cork and holm oak stands in Portugal, using a longer time range of 34 years. EVI could have been an interesting vegetation index for this analysis since it is less sensitive to understory spectral response. However, it has been proven to be significantly affected by differences in the sensor view geometry of Landsat satellites, as opposed to NDVI [28].

Since Landsat 4 satellite was launched in 1982, Landsat imagery allows producing long-term trends and spatial analysis at the fine resolution of 30 meters, facilitating detailed forest monitoring. A comparison of Landsat 30-m and MODIS 500-m-derived NDVI with flux tower data [29] revealed Landsat gives a vegetation phenological signal closer to the flux tower than MODIS. This approach implies to use more than one Landsat sensor. Several studies revealed disparities between the sensors, in particular between Landsat TM 5 and ETM+ 7 NDVI values that can introduce artificial long-term trends, due to differences in the bands wavelength sensibility and atmospheric conditions [28,30,31]. Those values need to be adjusted. As a consequence, our study will rely on inter-calibrated Landsat-derived NDVI time series, and will also use MODIS data to validate the adjustments performed. Phenological monitoring of both Californian *Quercus douglassii* savannas and pure grasslands areas with NDVI [32] showed phenological dates were consistent across spatial resolution for grasslands but varied in savannas. The conclusion of this study was that 500-m pixels give an average of the wide range of information that can be observed at a finer scale. Thus, we expect to find a difference in NDVI values between Landsat and MODIS time series, but a coherent trend direction, as long as the study-area used to compare them is large enough.

Stand crown cover in *montado* systems is commonly low, ranging from 10 to 50% [17]. A study estimating tree canopy cover in cork and holm oak stands using vegetation indices pointed out the necessity to use a period when the spectral contrast between the overstory and the understory is

maximum to focus on overstory signal [33]. This strategy was also applied for monitoring woody vegetation in Israel [34] and for adjusting the NDVI values of two Landsat sensors [35]. Moreover, a comparison of trees (*Q. suber*), grass and shrub reflectance over the years [36] has proven that the cork oak NDVI response was close to steady through the year, while the herbaceous vegetation reflectance shows large variations, due to phenology, being the lowest in July and August. Our time series will be based on those months.

To follow the long-term productivity trends of cork and holm oaks for a whole country, the potentialities of the new open access platform Google Earth Engine (GEE, [37]) were exploited. GEE gives free access to thousands of geographic information system features (vector and raster) from all over the world. Its JavaScript code editor allows a user to manipulate them online, without downloading, and to save images or table results on an online storage platform. Therefore, this work will be an opportunity to evaluate to what extent GEE is adapted to the needs and purpose of image processing with the goal of forest monitoring.

The aim of this study was to analyze the spatial and temporal trends of NDVI in cork and holm oaks forests of Portugal and to identify and quantify the areas with increasing and decreasing productivity trends. Here, we extended previous research by using (i) a long time series (34 years), (ii) a higher spatial resolution (30 m), and (iii) a contextual statistical technique to take advantage of the fine spatial resolution and account for neighboring information. Additionally, we selected six case studies located in cork and holm oak permanent forest inventory plots to further explore the relationship between NDVI trends and oak woodland productivity. As a secondary aim, we developed a flexible methodology of long-term monitoring based on remotely sensed data that can be applied to other types of forests and vegetation indices.

## 2. Datasets and Pre-Processing

The main steps of the methodology, summarized in Figure 1, were: (i) to define a study area including all unburned cork and holm oak areas of Portugal with a constant land cover class from 1984 to 2017; (ii) to identify six representative study sites in Portugal; (iii) to produce Landsat NDVI times series, using both TM and ETM+ imagery, which required an adjustment of NDVI values from one sensor to the other; (iv) to elaborate a MODIS NDVI time series to be compared with the Landsat time series in the study sites; (v) to determine NDVI long-term trends with Mann-Kendall (MK) and Contextual Mann-Kendall (CMK) tests, and their rate of change with the Theil-Sen slope estimator (TS).

### 2.1. Spatial Distribution of Cork and Holm Oaks in Portugal

The spatial distribution of cork and holm oaks was derived from the Land Use and Land Cover Maps of Portugal (*Cartas de Uso e Ocupação do Solo*—COS, [38]), which is currently available for 1995, 2007, 2010, and 2015. The COS characterizes homogeneous areas of at least 1 hectare with a five-level hierarchical classification (193 classes for the most detailed level). The most common cork and holm oak stand cultural systems were summarized into five classes, defined following COS class descriptions [39]:

1. Cork oak forests: The tree canopy covers no less than 10% of the area (for 4-meter canopy radius, 20 trees/ha). The understory is not used for agriculture (agroforestry) or recreative activities (urban parks). This class regroups pure and mixed forests (hosting other broadleaf and coniferous species) dominated by cork oaks;
2. Holm oak forests: Forests as defined above for cork oaks, in this case dominated by holm oaks;
3. Cork oak agroforestry systems (AFS): AFS are associations of temporary crops or grasslands with permanent tree species. At least 10% of the area is covered by the forest canopy and dominated by cork oaks;
4. Holm oak agroforestry systems: AFS, as defined above, here dominated by holm oaks;

5.  Agroforestry systems with cork and holm oaks: Particular case of AFS, as defined above, in which both species co-occur and none represents more than 75% of the forest canopy cover.

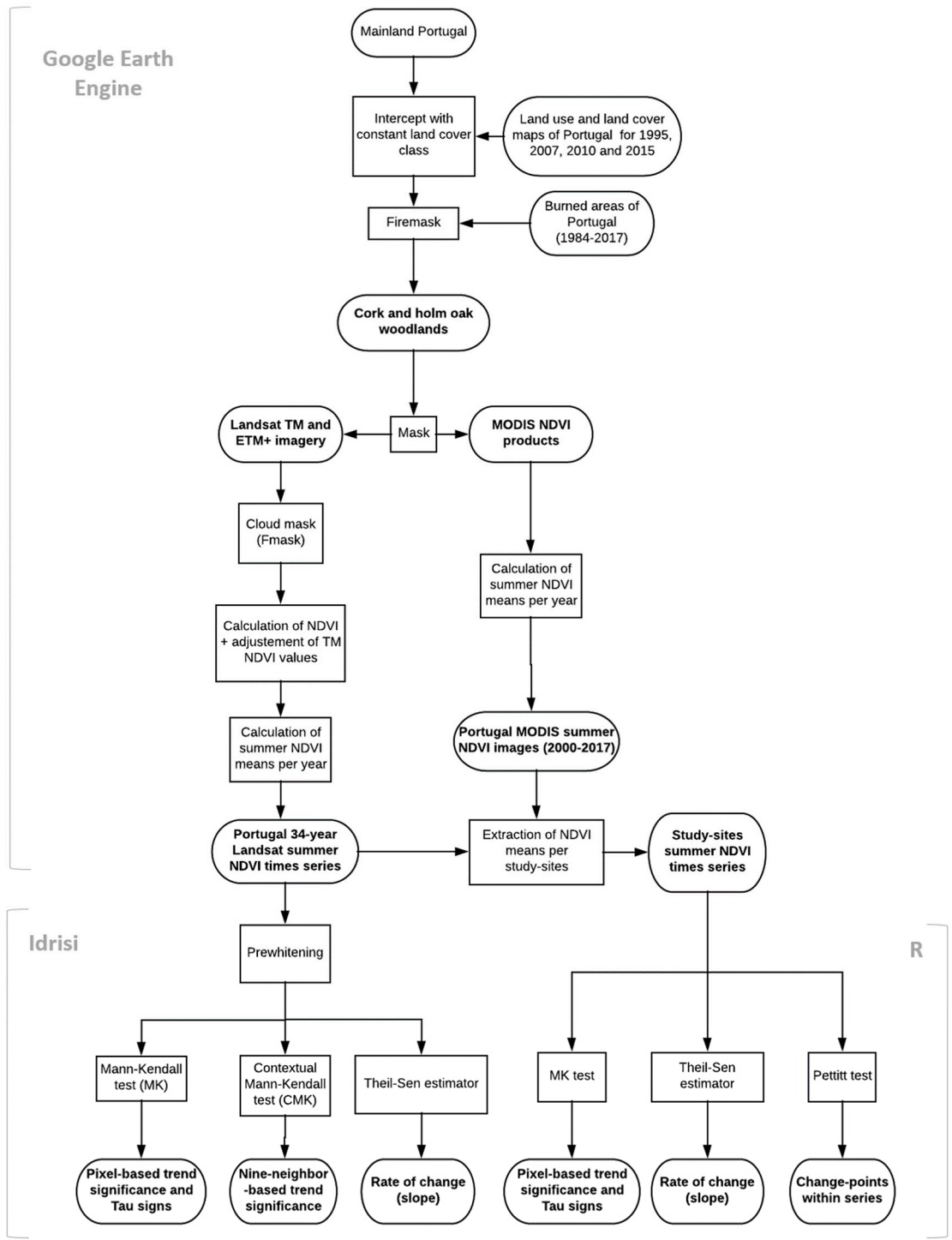

**Figure 1.** Flowchart of the methodology, allowing to build long-term NDVI time series (top, in Google Earth Engine), to compare MK and CMK results at Portugal scale (bottom left, in Idrisi) as well as Landsat and MODIS NDVI results for the six study sites (bottom right, in R).

Since our goal was the monitoring of cork and holm oak stands and not the study of transitions from or to oak land cover classes, only the areas with a constant land cover class, i.e., areas where class transitions were not observed in the period 1995–2015, were considered in our analysis. No COS data

exists to guarantee the land cover of those areas remained constant between 1984 and 1995, as well as for the period 2015–2017. New plantations were important in the 1990s due to financial programs and could produce an over-estimation of positive NDVI trends. This is anyway reflecting the global evolution of cork and holm oaks in Portugal. Most of all, the stands covered by cork and holm oaks are very stable over time, due to their type of exploitation and potential lifetime and since those protected species cannot be cut without formal permission, according to the current national legislation in Portugal. Using the available COS, it was possible to estimate that the land cover changes from 1995–2015 were mainly new plantations (35%) and conversion between a forest and an agroforestry system of the same species (31%). Many areas corresponded also to conversions from one oak species to another, always remaining in the five classes we defined (24%). The last 10% corresponded to stands replaced by urban areas, agriculture, shrublands or other woody species or to abrupt transitions (absence for 3–8 years), possibly due to fires and replantation. With those limits in mind, it was assumed the stands with a stable land cover class from 1995 to 2015 were suitable for a 1984–2017 trend analysis (Figure 2A).

Areas burned between 1984 and 2017 (Figure 2B, in grey) were used as a mask to exclude from the analysis the burned stands and again to focus on the areas that remained covered by cork or holm oaks. Burned area perimeters have been extracted from the Landsat-based Fire Atlas of Portugal [40], produced by the Forest Research Centre (School of Agriculture, University of Lisbon). This dataset provides burned area polygons over the entire mainland Portugal for each fire season between 1975 and 2013. It was produced with a semi-automatic classification of Landsat satellite imagery (Landsat TM/ETM+ era) and has a minimum mapping unit of 5 hectares. Data from 1984 to 2013 were merged with annual burned areas maps from the Institute for Nature Conservation and Forests [41] for the 2014–2017 period.

Cork and holm oak stands are not situated in the regions of Portugal most affected by fire (Figure 2), mostly located in the northern half of the country. Only the most southern region of Portugal, Algarve, as defined in the Nomenclature of Territorial Units for Statistics level 2 (NUTS II) [42], has large areas of oak stands and presents a high fire incidence. Table 1 summarizes the area covered by cork and holm oaks, before and after removing the burned areas, and the proportion of burned area removed per land cover class. Close to 13% of the stable oak stands were masked, mainly from cork and holm oak forest classes. The total area finally used in our analysis covers 1,902,360 ha.

**Table 1.** Area covered by each land cover class concerning cork and holm oaks in Portugal before and after fire masking and the proportion of the area removed.

| Land Cover Class | Initial Area (ha) | Fire-Masked Area (ha) | Proportion of Area Removed (%) |
|---|---|---|---|
| Cork oak forests | 844,153 | 663,644 | 21.4 |
| Holm oak forests | 264,896 | 225,879 | 14.7 |
| Cork oak agroforestry systems | 292,334 | 273,598 | 6.4 |
| Holm oak agroforestry systems | 614,577 | 586,124 | 4.6 |
| Agroforestry systems with cork oak and holm oak | 161,158 | 153,114 | 5.0 |
| **Total** | 2,177,117 | 1,902,360 | 12.6 |

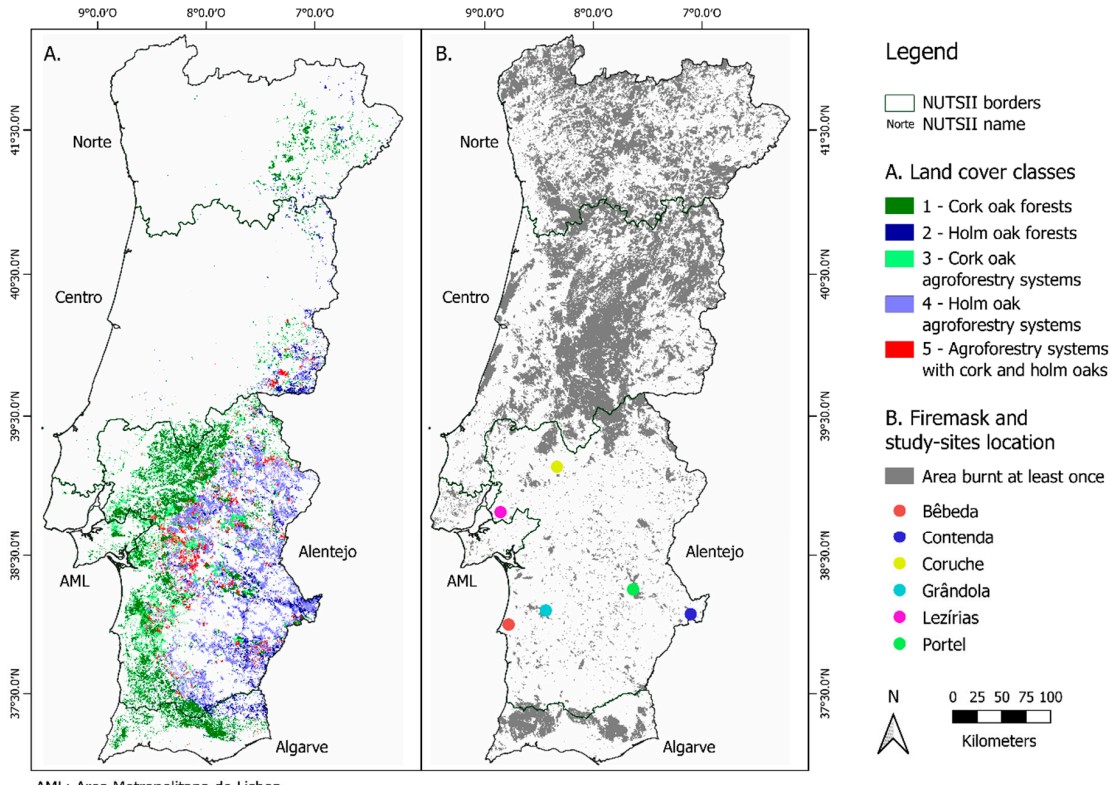

AML: Area Metropolitana de Lisboa

**Figure 2.** Study area, consisting of cork and holm oak areas with a constant land cover class in the period 1995–2015, according to the Land Use and Land Cover Maps of Portugal (**A**); burned areas in mainland Portugal (from 1984 to 2017), which were removed from the analysis, with the locations of the six study sites (**B**).

## 2.2. Representative Study Sites

Six representative cork oak stands were selected to help to choose the best adjustment between TM and ETM+ NDVI values and to explore the relationship between observed NDVI trends and field productivity. Those sites contrast in their density, understory composition, cork production, debarking history, and climate, following a coast-inland gradient (Figure 2A).

Bêbeda plot (38°0′2.7″ N, 8°46′52.0″ W), situated on Arenosols [43], 6 km from the sea, is a stand of 70 years old with 13.1% canopy cover (according to the 30-m pixel Landsat Tree Cover Continuous Fields layer for 2010, which gives a percentage of the vertically projected area of vegetation of woody plants greater than 5 meters in height [44], although those values seem very low compared to field estimations (personal communication)). Contenda stand (38°4′32.6″ N, 7°6′14.2″ W) is an uneven-aged plot of 100 years old or more with natural regeneration on Leptosols, with only 8.8% canopy cover. It is the farthest study site from the sea (150 km), with a drier climate than the other stands (its mean precipitation per year is 450 mm against around 550 mm) but with similar Köppen *Csa* characteristics, especially a wet winter (around 55 mm of rain in December and around 75 for the other stands) and dry and hot summer months (less than 5mm of rain in average in July for all plots). Coruche (39°8′21.0″ N, 8°20′8.1″ W) is the youngest stand, about 60 years old, and has 18.8% canopy cover (estimated at 40% with unmanned aircraft vehicle image mosaic; study in preparation), hosting a cow pasture on its Podzols soil 90 km from the sea. Grândola (38°6′8.1″ N, 8°26′18.8″ W), at 30 km from the sea, is an uneven-aged plot of more than 100 years old hosting a sheep pasture on Arenosols and Podzols under only 7.3% canopy cover. Lezírias (38°48′55.7″ N, 8°51′20.1″ W) is a 70-year-old plot on Arenosols 40 km of the sea, with 23.3% canopy cover, the highest proportion of the six study sites. At last, Portel (38°15′14.2″ N, 7°38′3.1″ W) is a 100-year-old plot with 12.8% of canopy cover 100 km from the sea, hosting sheep and cows on Leptosols.

The cork productivity of those plots has been followed since the 1990s by the Forest Ecosystem Management under the Global Change (ForChange) research group of the Forest Research Centre. The area of each plot is around 1 hectare, where it fits 11 Landsat pixels. The plot center was selected in order to get the most homogeneous canopy cover possible.

### 2.3. Landsat and MODIS Imagery

Landsat imagery was selected because they allow a long-term time series analysis at a high spatial resolution. Landsat-5 Thematic Mapper (TM) and Landsat-7 Enhanced Thematic Mapper Plus (ETM+) data were used to create a 34-year time series from 1 January 1984 to 31 December 2017 with a 30-m spatial resolution and a 16-day revisiting period. The NDVI (Equation (1)) [45] was calculated with the Surface Reflectance (SR) images provided by GEE:

$$\text{NDVI} = (\text{NIR} - \text{Red})/(\text{NIR} + \text{Red}), \tag{1}$$

where NIR and Red are TM or ETM+ Near Infrared and Red bands, respectively. Only the first category of data (Tier 1: "T1_SR") produced by the United States Geological Survey (USGS) was used (Table 2). According to the USGS Landsat, scenes with the highest available data quality are placed into Tier 1 and are considered suitable for time-series processing analysis. A Fmask function [46] was implemented in GEE to remove clouds and shadows from Landsat images, using the quality of pixel bands (pixel_qa). Moreover, all images of ETM+, taken after 31 May 2003, contain "SLC-off gaps" due to the failure of the Scan Line Corrector (SLC) of the sensor. On GEE, the values of the missing pixels are classified as 'NA' values. No interpolation to fill those gaps was made.

**Table 2.** Image collections used, available in Google Earth Engine (GEE).

| Sensor | Range of Dates | Time Scale | Spatial Resolution | GEE Image Collection ID |
|---|---|---|---|---|
| Landsat-5 TM | 1984-01-01–2012-05-05 | 16-day cycle | $30 \times 30$ m | LANDSAT/LT05/C01/T1_SR |
| Landsat-7 ETM+ | 1999-01-01–2017-12-31 | 16-day cycle | $30 \times 30$ m | LANDSAT/LE07/C01/T1_SR |
| MODIS Terra V6 Vegetation Indices | 2000-02-18–2017-12-31 | 16-day cycle | $250 \times 250$ m | MODIS/006/MOD13Q1 |

As referred to in the Introduction section, although USGS considers all Tier 1 Landsat data radiometrically calibrated and geolocated consistently across the full collection for all the sensors, several studies revealed differences between the values of NDVI due to differences in the bands wavelength sensibility between TM and ETM+ [28,31] or atmospheric conditions [30]. Some linear relations between NDVI values from ETM+ and TM sensors have already been established for Northern Europe [47] (Equation (2)):

$$\text{ETM+} = 1.0210 * \text{TM} - 0.0010, \tag{2}$$

and more recently for Canada [35] (Equation (3)):

$$\text{ETM+} = 1.0370 * \text{TM}. \tag{3}$$

To compare site-specific calibration factors with those retrieved from the literature review, site-specific equations were derived from the six study sites described above, by adapting the methodology of tandem images cross-calibration [31] to the Landsat images available in Portugal. For each site, NDVI values were extracted from 1 January 1999 to 5 May 2012, the period shared by the two sensors, for each pixel of each site. Site-specific relations were assumed to be linear, with an intercept equal to zero, given that the literature equations' intercepts were very small or null (Equations (2) and (3)). Only image values from different sensors separated by a time-gap of 8 days (minimum gap possible for Portugal) were used, backward and forward. NDVI adjustments derived from each study site were compared with those obtained from Equations (2) and (3). Since the populations do not follow a normal distribution and time series observations are auto-correlated by

nature, the percentages of mean and median differences between raw and adjusted NDVI values were compared to determine the best relation to use for Portugal oak areas.

To assess the effect of using coarse spatial resolution satellite data, which would be more suitable for monitoring large study areas, the trends derived from Landsat were compared with those derived from MODIS imagery. We have selected the version 6 of the MODIS Terra Vegetation Indices (Table 2), which provides the NDVI at 250-m spatial resolution every 16 days between 2000 and 2017, already fully corrected and cloud-masked.

## 2.4. Trend Analysis

Under the hypothesis that the trends were monotonic, both Mann-Kendall (MK) [48,49] and Contextual Mann-Kendall tests (CMK) [50] were run in IDRISI 17.0 Selva Edition to obtain the strength and direction of the trends. The original pixel-based non-parametric test of MK, robust to outliers, determines if a time series has no monotonic trend, either uneven or no clear trend (null hypothesis). The alternative hypothesis is the existence of a significant monotonic trend ($p$-value < 0.05). The sign of the test's statistics Tau, positive or negative, indicates respectively increasing or decreasing trends. The CMK test is a geostatistical object-based approach of the MK test, which takes into account the behavior of the first order neighbors to judge the significance of a pixel monotonic trend. It was assumed that the serial autocorrelation was of an order of one in the 34-year time series so obtained, most of which can be removed using the Durbin-Watson statistic [50]. The time series was thus first pre-whitened using Durbin-Watson residuals series pre-process of the IDRISI Earth Trends Modeler. A value of the trend rate of change (slope) was assessed by the Theil-Sen robust linear estimator (TS) [51,52]. The expected results were maps of monotonic trend significance obtained with both MK and CMK tests and a map showing the TS slope estimate of those trends.

Based on previous works exposed in the Introduction section regarding intra-annual variations of NDVI in woodlands, only July and August NDVI values were used for the trend analysis. This allows to minimize the spectral contribution of understory vegetation and its phenological variability, and in this way, focus on the tree layer. Moreover, Ju et al. already used exclusively those 2 months to obtain better linear relations between TM and ETM+ NDVI values [35].

A similar analysis was made for the six study sites, to compare Landsat and MODIS NDVI values and trends, validate Portugal trend map results and present an overview of stand monitoring. Trends of 1-ha average summer NDVI were estimated with the MK test and the TS estimator for MODIS and Landsat imagery. Moreover, the non-parametric Pettitt test [53] was used to approximate the year of the significant change-point within the site-study trends. Those tests were run in R software using packages *stats* version 3.4.2 [54], *zyp* version 0.10-1 [55], and *trend* version 1.1.0 [56].

## 2.5. Exploring Relationships with Field Productivity

The study sites were an opportunity to verify if the NDVI trends were coherent with the field data. The current legislation in Portugal allows stripping trees with a minimum gap of 9 years. After the not-usable first two strippings, the workable cork is extracted from 50 to 150-year-old trees or more. The cork production of the study sites had been measured for two consecutive debarking periods (a total of 17 to 22 years depending on the study site), within the 1985–2012 period. Annual rings thickness was measured after boiling cork samples [18,57]. The first idea was to compare cork production and NDVI trends. However, the cork production is higher on the year following the debarking, due to the induced hydric stress, and decreases then regularly until the next cork harvest [58]. The response of the trees depends on their own phloem tissues composition as well as on the intensity of the stripping [59], with the last varying with site conditions [60]. Only cork samples from living trees had been followed in each plot, introducing a bias in the plot's actual health. Thus, to validate NDVI trend results with field data, it was chosen to look only at the global productive state of each stand, namely the average production of cork per tree per year. The average was calculated for each stand from samples of 23 to 36 trees and compared with the NDVI TS slope, which is judged

a better indicator of the productivity state of the stands and its evolution than NDVI means, which depends mainly on the canopy cover [61], and thus, on the density of trees more than their health state.

## 3. Results

### 3.1. TM and ETM Adjustment

Between 1165 and 6445 values were used for each site, depending on the cloud cover, which was larger on the sites closer to the sea. A strong linear relationship between TM and ETM+ NDVI values was found for each site, with the coefficients of determination ($R^2$) ranging from 0.9754 to 0.9922 (Table 3).

**Table 3.** Site-specific linear relations between ETM+ and TM NDVI values.

| Location | Equation | $R^2$ |
|---|---|---|
| Bêbeda | ETM+ = 1.0205 * TM | 0.9866 |
| Contenda | ETM+ = 1.0101 * TM | 0.9754 |
| Coruche | ETM+ = 1.0124 * TM | 0.9922 |
| Grândola | ETM+ = 1.0109 * TM | 0.9821 |
| Lezírias | ETM+ = 1.0039 * TM | 0.9878 |
| Portel | ETM+ = 1.0345 * TM | 0.9772 |

Table 4 presents the proportions of difference between raw or adjusted TM and ETM+ NDVI values. The adjustments give similar results. Given that using a unique equation is much more efficient in terms of processing, especially for a whole country, we selected the one presenting the minimal average error for the six study sites: Equation (3).

**Table 4.** Comparison of TM and ETM+ NDVI values for each study site: proportion of mean and median errors using raw data and three different adjustments.

| Site | Type | Raw Data | Equation (2) | Equation (3) | Site Specific |
|---|---|---|---|---|---|
| Bêbeda | Mean error (%) | −5.275 | −3.492 | −1.770 | −3.329 |
| Contenda | Mean error (%) | −8.785 | −7.085 | −5.410 | −7.868 |
| Coruche | Mean error (%) | −3.499 | −1.644 | 0.072 | −2.303 |
| Grândola | Mean error (%) | −9.750 | −8.074 | −6.411 | −8.764 |
| Lezírias | Mean error (%) | −1.707 | 0.177 | 1.930 | −1.324 |
| Portel | Mean error (%) | −7.582 | −5.880 | −4.162 | −4.391 |
| **Average mean error (%)** | | **−6.099** | **−4.392** | **−3.292** | **−4.663** |
| Bêbeda | Median error (%) | −5.010 | −3.224 | −1.496 | −3.060 |
| Contenda | Median error (%) | −13.134 | −11.540 | −9.920 | −12.260 |
| Coruche | Median error (%) | −3.293 | −1.435 | 0.285 | −2.094 |
| Grândola | Median error (%) | −10.412 | −8.772 | −7.097 | −9.433 |
| Lezírias | Median error (%) | −1.815 | 0.066 | 1.818 | −1.432 |
| Portel | Median error (%) | −6.667 | −4.963 | −3.213 | −3.444 |
| **Average median error (%)** | | **−6.722** | **−5.000** | **−3.971** | **−5.287** |

### 3.2. Country Level Cork and Holm Oak NDVI Trends

Figure 3 presents the maps of NDVI evolution for all significant trend areas following the CMK test. Figure 3A shows the sign of CMK's Tau, with positive values characterizing increasing trends (in blue) and negative values decreasing trends (in red). The statistics of this map are summarized in Table 5: 70.65% of CMK significant areas were found increasing and 29.35% decreasing, which accounts for 28.36% and 11.78% of the total area of oak forests. These results show that more than one-tenth of the area of cork and holm oak stands in mainland Portugal have significant declining trends over the 34-year period. The most affected areas are located in the northern, coastal and central parts of the Alentejo NUTS II region, where decreasing clusters are located (Figure 3A). Conclusions

regarding the Algarve region have to be moderated since most of its oak stands have suffered forest fires these last few decades and have been masked in this study (Figure 2B).

Figure 3B gives the annual rate of change of NDVI, obtained by the TS slope estimator, for the 34 years of the time series, for the areas with significant trends (Figure 3A). Rates values followed a normal distribution. For CMK significant trend pixels, 95% of the values were between −0.0065 and 0.0059. The highest negative values were found in Serra de Portel and in the south of Serra de Grândola.

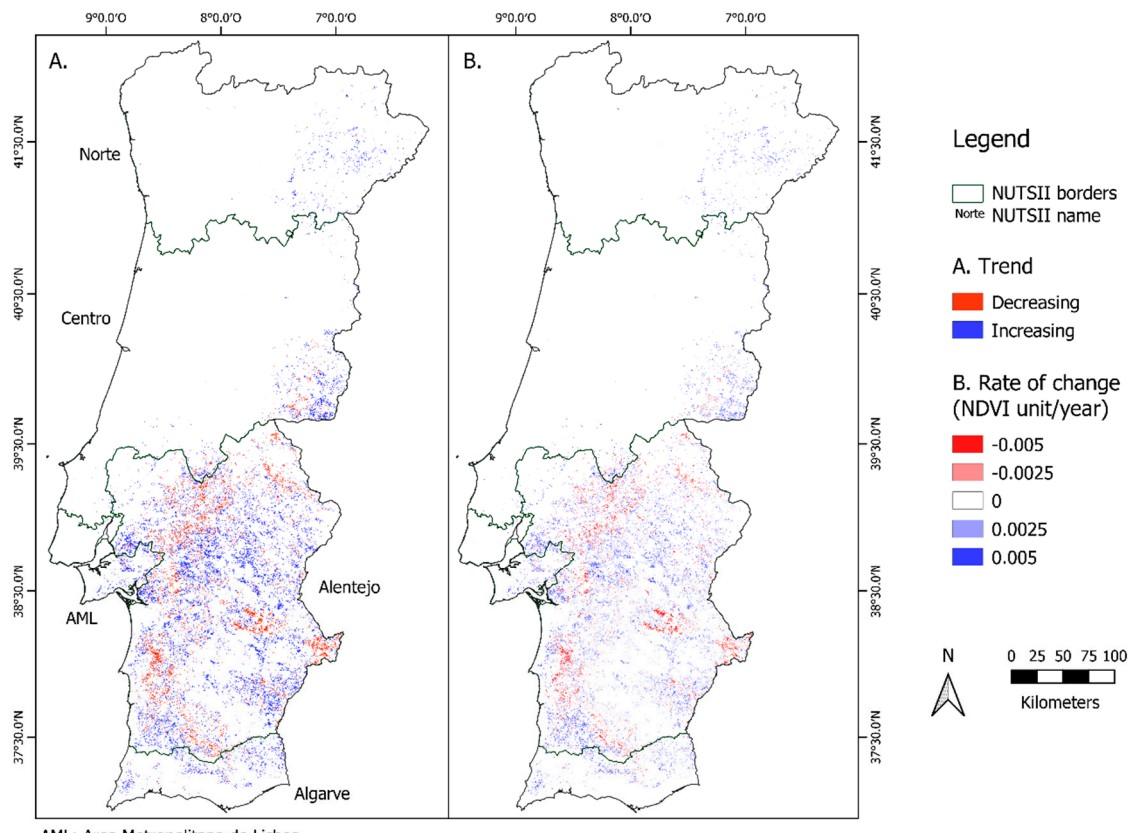

AML: Area Metropolitana de Lisboa

**Figure 3.** NDVI trends for cork and holm oak areas in Portugal for the period 1984–2017, with (**A**) significant decreasing (in red) and increasing trends (in blue) based on the Contextual Mann-Kendall test; and (**B**) the NDVI rate of change given by TS estimator.

**Table 5.** Area and proportion of the cork and holm oak total area found for NDVI trends in Portugal, obtained with the Contextual Mann-Kendall test.

|  | Area (ha) | Proportion of Total Area (%) |
|---|---|---|
| Significant trends of which: | 763,573 | 40.14 |
| - increasing trends | 539,451 | 28.36 |
| - decreasing trends | 224,109 | 11.78 |
| Not significant trends | 1,138,787 | 59.86 |
| **Total area** | **1,902,360** | **100** |

CMK significant increasing and decreasing trends for each land cover class are compared in Figure 4. The proportion of significant trends by land over class varied between 35% for holm oak agroforestry systems and 47% for holm oak forests. Cork oaks agroforestry systems and forests had respectively 41% and 43%, while agroforestry systems with both species presented 36% of the significant trend areas.

According to the maps presented in Figure 4, cork oak agroforestry (C) appears to be in steeper decline in land cover (16.66% of the area of this class presents a significantly decreasing trend), while holm oak agroforestry systems (D) present less than half of this value (7.68% of the class total area). Agroforestry systems with both species (E) are in an average state between those two values (12.49%). Forests of cork oak (A) and holm oak (B) have close values, respectively 13.13% and 12.06% of their areas reveal decreasing significant trends. When several classes coexist, trend values are coherent. The main decreasing NDVI trends are found in the same locations whatever the cultural systems: for example, in the south of Grândola for cork oak forests and agroforestry systems (A and C); around Portel for holm oaks (B and D). Forests of cork and holm oaks (A and B) evolve the same way where they co-occur, both presenting downward trends around Portel and Grândola and upward trends in the east of Castelo Branco.

*3.3. Study Sites Trends*

Figure 5 presents the trend results at a detailed spatial scale, in windows of 3 km$^2$ around each of the six study sites. Figure 5A shows the significant trend areas (black polygons) according to MK and CMK tests. The pixel-based MK test has a noisy result, with more isolated significant pixels that are ignored with the contextual approach (CMK), while the CMK test produces larger, more homogeneous clusters of pixels with significant trends. Those differences were expected and have already been discussed in CMK test evaluation works [50]. The CMK approach seems more representative and more suitable to apply at 30-meter spatial resolution.

It is possible to notice some large patches of pixels with the same trend (e.g., Bêbeda, Portel), which may correspond to a unique forest stand, submitted to the same pedoclimatic conditions and the same management options (age, density). On the contrary, clusters of opposite significant trends can be observed side by side, for example, around Coruche site. The detailed shape and size of those small units clearly demonstrates the interest of using 30-m spatial resolution satellites images.

The TS slope estimator (Figure 5B) gives the magnitude of the trends, which is useful to unveil the different behaviors of the stands (for example, Lezírias and Grândola show both increasing trends, but with a very different rate of change) or identify peculiar patterns (discernible roads around Bêbeda) for precise monitoring purposes.

Portel stand is not located inside a cluster on the trend maps because it does not have a constant land cover class from 1995 to 2015. A different delimitation of COS patches between 1995 and 2007, and possibly shrub encroachment, made it appear first as cork oak agroforestry systems with pasture, then as cork oak forest.

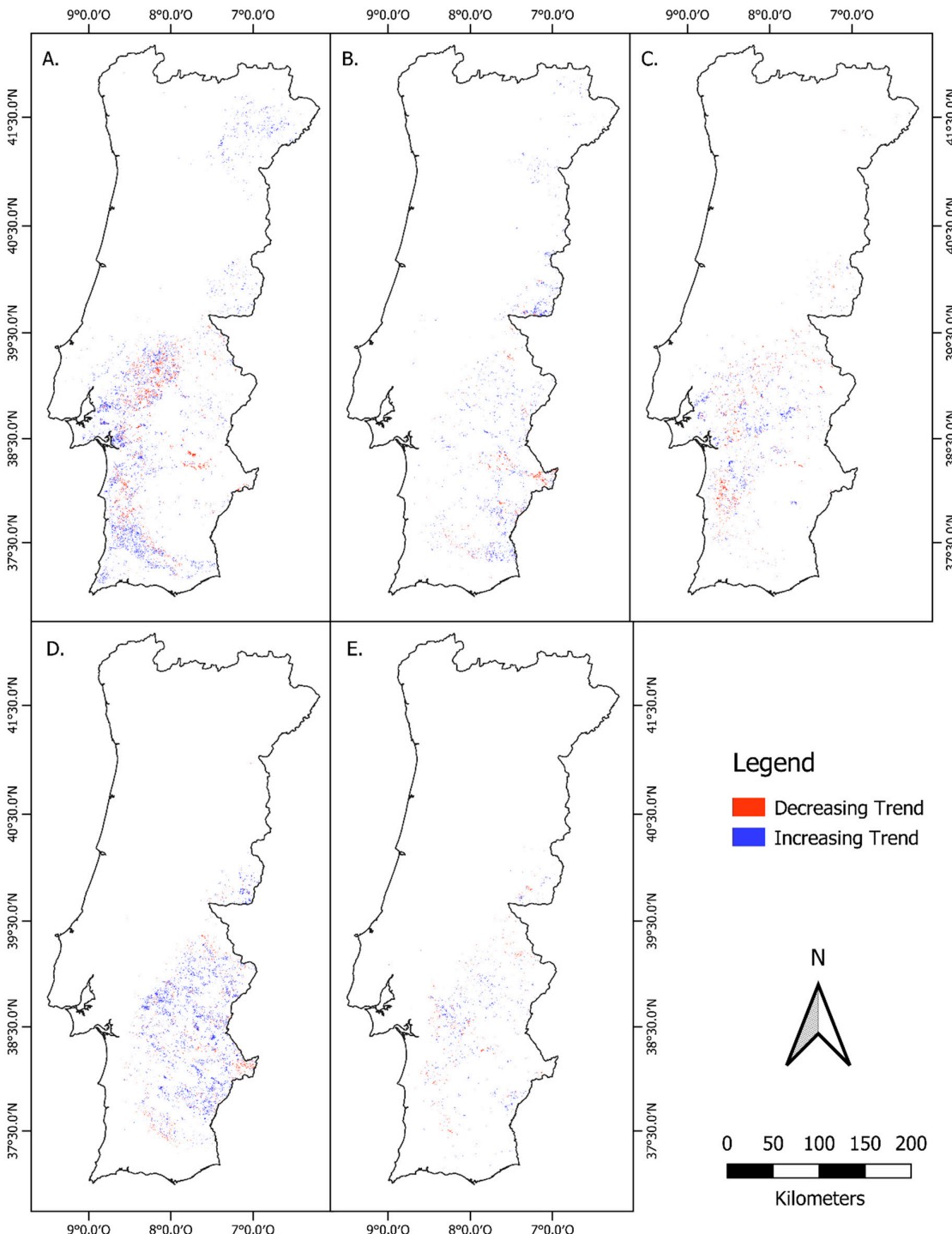

**Figure 4.** Significant NDVI trends for cork and holm oak areas in Portugal (1984–2017), based on the Contextual Mann-Kendall test, per land cover class: (**A**) cork oak forests; (**B**) holm oak forests; (**C**) cork oak agroforestry systems; (**D**) holm oak agroforestry systems; and (**E**) agroforestry systems with cork oak and holm oak.

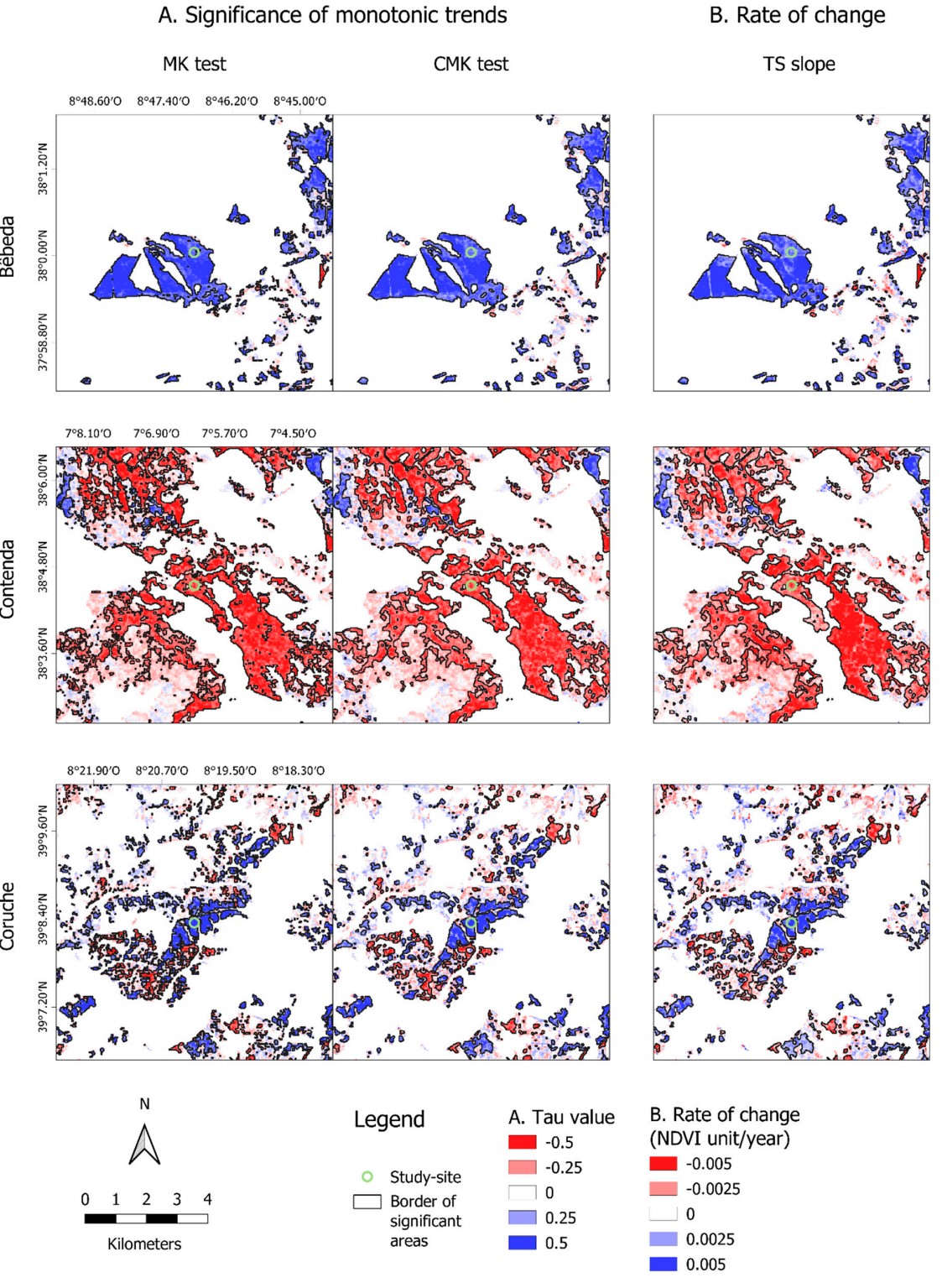

**Figure 5.** *Cont.*

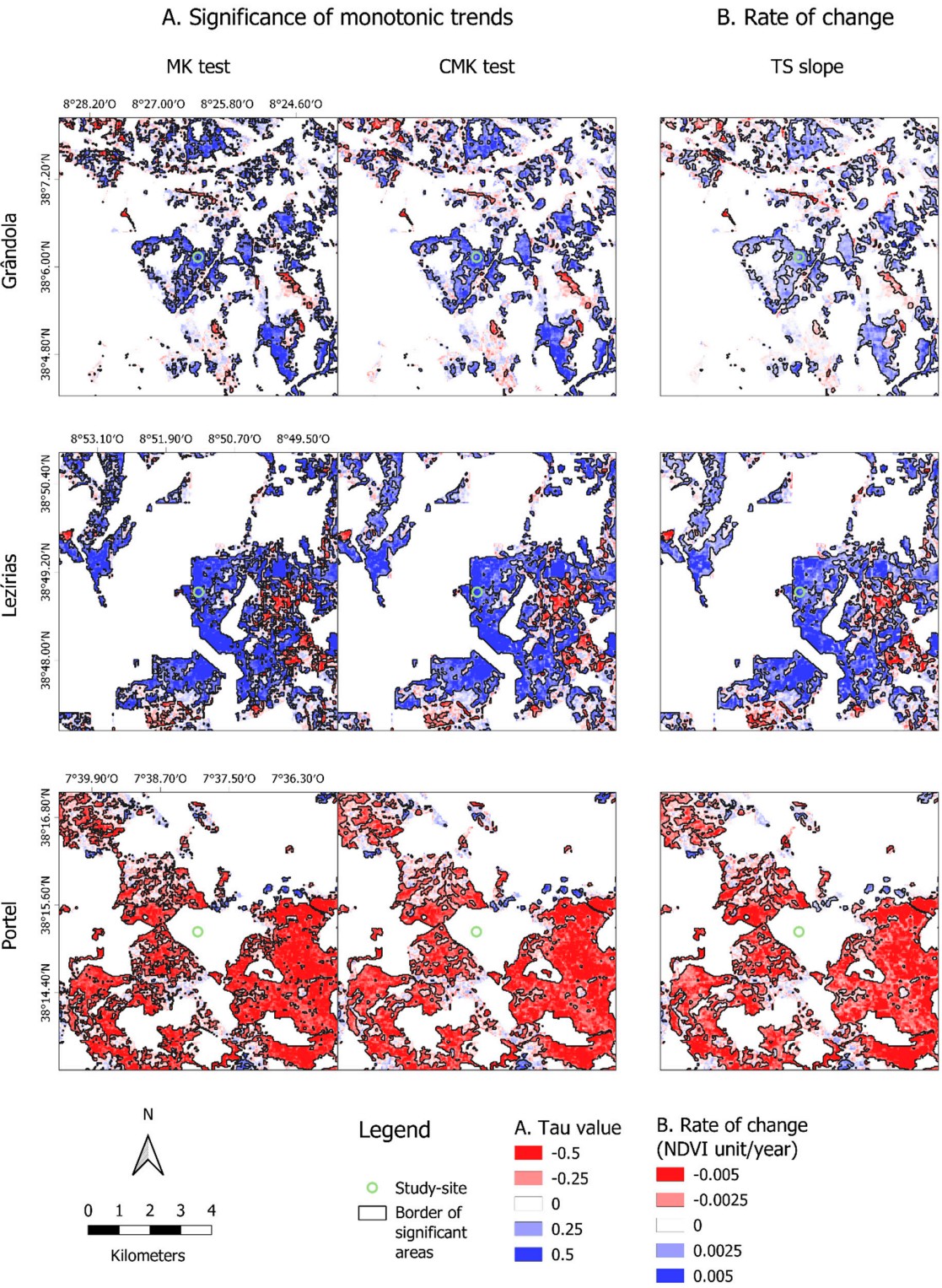

**Figure 5.** Details around the six study sites: (**A**) comparison of significant NDVI trend areas (black polygons) determined by MK (left) and CMK tests (right), Tau values allow to visualize increasing (in blue) and decreasing (in red) trends; (**B**) rate of change of NDVI values (TS slope in NDVIunit.year$^{-1}$), from red (lowest) to blue color (highest), with the limits of CMK significant trend areas overlaid (in black).

### 3.4. Comparison with MODIS Trends

To validate the adjustment of TM and ETM+ NDVI, Landsat trends were compared with the ones obtained with MODIS imagery. The Mann-Kendall test and the Theil-Sen estimate were applied to the July–August average NDVI time series of each study site. Landsat results are presented for the whole period (1984–2017) and for the 17 years (2000–2017), the most common period with the MODIS sensor.

Table 6 presents the Tau and p-value obtained with the Mann-Kendall test. All Landsat long-term trends (1984–2017) were found to be significant with the MK test (p-value < 0.05). Considering the last 17 years (2000-2017), significant monotonic trends appeared in Coruche, Grândola and Lezírias stands with Landsat imagery, and in Grândola and Lezírias using MODIS NDVI. The shorter duration of the time series may explain the lack of significance in some sites. Tau signs were consistent for every site between the two Landsat and MODIS time series, except for Bêbeda.

**Table 6.** Comparison of Landsat and MODIS Mann-Kendall test results for each study site.

| Study Site | Landsat (1984–2017) | | Landsat (2000–2017) | | MODIS (2000–2017) | |
|---|---|---|---|---|---|---|
| | Tau | *p*-Value | Tau | *p*-Value | Tau | *p*-Value |
| Bêbeda | 0.4581 | $8.64 \times 10^{-5}$ | −0.0065 | 1.0000 | 0.0327 | 0.8814 |
| Contenda | −0.4831 | $3.14 \times 10^{-5}$ | −0.2288 | 0.2008 | −0.0525 | 0.7617 |
| Coruche | 0.6542 | $2.96 \times 10^{-9}$ | 0.3464 | 0.0477 | 0.1242 | 0.5009 |
| Grândola | 0.4759 | $4.23 \times 10^{-5}$ | 0.4771 | 0.0051 | 0.5948 | 0.0003 |
| Lezírias | 0.6054 | $4.99 \times 10^{-7}$ | 0.5508 | 0.0015 | 0.4459 | 0.0100 |
| Portel | −0.6364 | $9.83 \times 10^{-9}$ | −0.2026 | 0.2599 | −0.2288 | 0.2008 |

Rates of change obtained with TS linear slope estimator, expressed in NDVIunit.year$^{-1}$, are presented in Table 7. Regarding the 34-year period, four stands presented increasing trends and two were significantly decreasing, Contenda and Portel. The most contrasting slope values were found in Portel (-0.0051) and Coruche plots (0.0059). Those results are in accordance with the rates of change observed in each plot map (Figure 5B). Landsat slope values are quite different for the two periods considered. For the 17-year trend, they appear either much lower (Contenda, Portel) or higher (Grândola, Lezírias) than 34-year slopes.

**Table 7.** Comparison of Landsat and MODIS Theil-Sen linear slope estimate for each study site.

| Study Site | Landsat (1984–2017) | Landsat (2000–2017) | MODIS (2000–2017) |
|---|---|---|---|
| Bêbeda | 0.0034 * | −0.00025 | 0.0002 |
| Contenda | −0.0042 * | −0.00199 | −0.0005 |
| Coruche | 0.0059 * | 0.00339 * | 0.001 |
| Grândola | 0.002 * | 0.00284 * | 0.0043 * |
| Lezírias | 0.0043 * | 0.00478 * | 0.0034 * |
| Portel | −0.0051 * | −0.00148 | −0.0014 |

Significant trends according to Mann-Kendall test (*p*-value < 0.05) are marked with an asterisk (*).

The non-parametric Pettitt change-point test was run for each study site of Landsat trends to detect potential change-points. Figure 6 presents the complete time series of Landsat and MODIS (respectively in blue and black) for each study site, the results of the TS linear estimator (blue lines) and Pettit test (vertical dash lines) for Landsat 34-year trends. The six study sites had a significant change-point of Landsat trend in the same decade, between 1996 and 2005. As seen in Figure 6, NDVI values follow similar annual variations for the two sensors and present coherent trends. However, the magnitude of trends, i.e., the TS slopes (Table 7), reveal some differences that may be a direct consequence of sensors disparities in spatial and spectral resolutions.

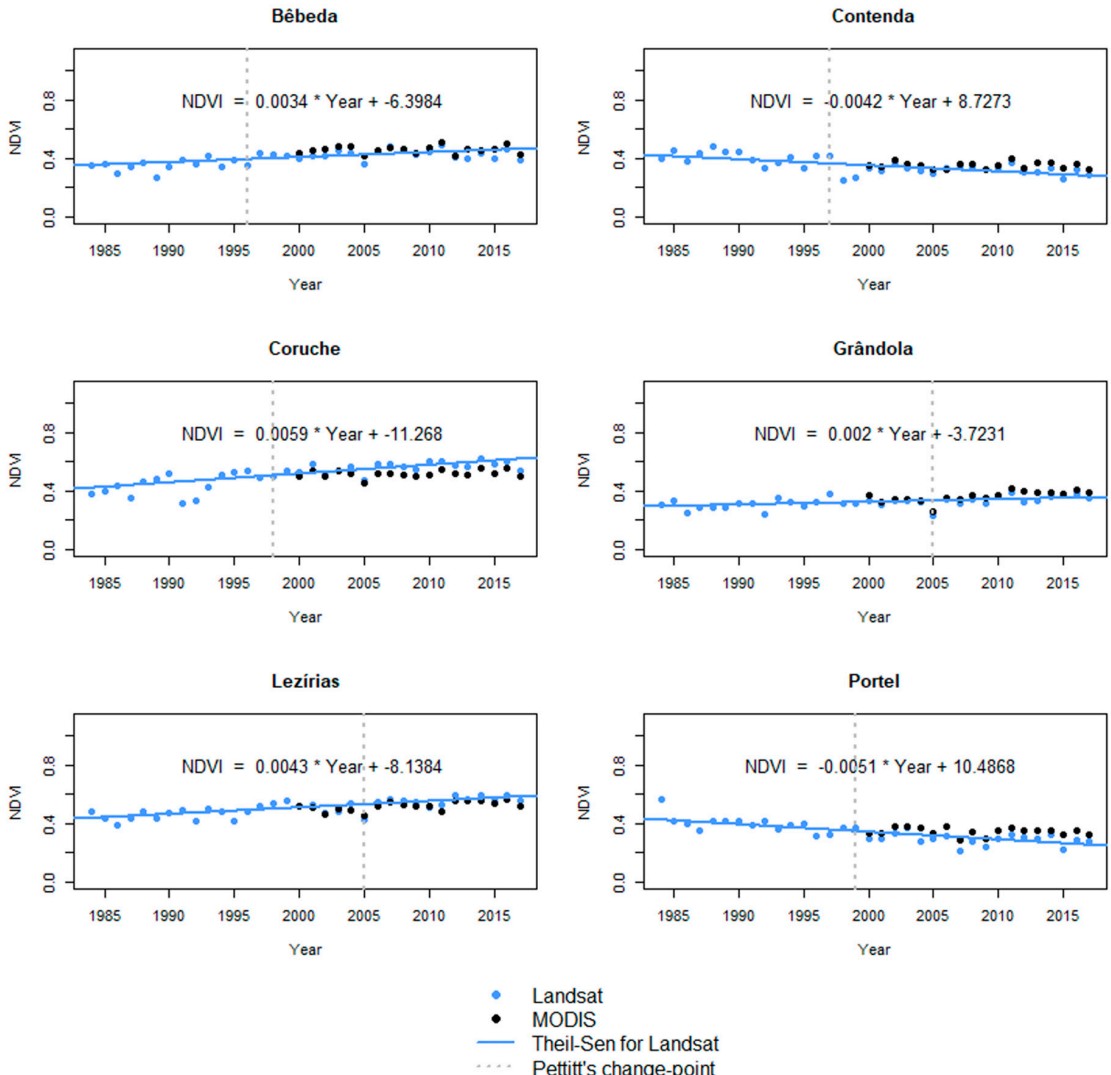

**Figure 6.** July–August times series per study site for Landsat 34-year trends (blue) and MODIS (black)-derived NDVI values, with the linear estimation of Theil-Sen and the Pettitt's change point of Landsat significant trends.

### 3.5. Comparison with Field Data

The mean cork production of each study site (see Section 2.5) was graphically represented as a function of the NDVI TS slope derived from the Landsat imagery (Figure 7). The two different types of data show a large agreement. Portel and Contenda stands, plots of 100 years or more, present low cork production and a clear NDVI decline, which can be the consequence of a reduction either in the number of trees or in their global health. On the contrary, Coruche plot, the youngest, stands out from the others with its high values of productivity and large NDVI increase. Lezírias, Bêbeda and Grândola sites, all on Arenosols, present similar productivities and NDVI trends despite their different proportions of canopy cover. Figure 7 suggests a linear or exponential relation between the two variables, although this study would need to be extended to more sites to strengthen these conclusions.

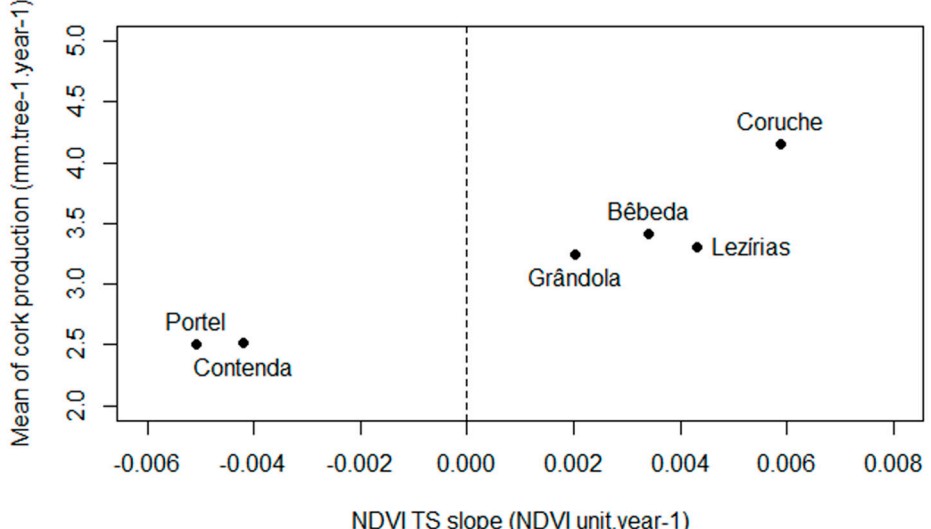

**Figure 7.** Twenty-year mean cork production in millimeters per tree per year as a function of 34-year Landsat NDVI trends for the six study sites.

## 4. Discussion

Our study aimed at mapping and estimating the direction and magnitude of change in the health and productivity of cork and holm oaks stands in continental Portugal. The methodology was successfully applied to map 34-year NDVI trends of all stands of cork and holm oaks. The 30-m resolution of Landsat images allowed an accurate identification of spatial patterns within the oak forests, efficiently avoiding roads, water surfaces, and adjacent croplands, without its processing being over-time-consuming in GEE.

Monotonic NDVI trends were calculated for 34 consecutive years, which was never done before, especially for the whole country. The comparison of MODIS and Landsat trends for 17 years showed the importance of using long time series, which ensures that the significant trends found are not due to short-term events such as droughts or debarking, but reflect a long-term evolution of the health of the forest stands.

Summer NDVI revealed to be a good proxy of cork oak stands health and productivity for the six study sites, presenting a similar evolution to the one expected regarding field data: the results suggested a good relation between summer NDVI trends and cork growth (Figure 7). The six study sites had a significant change-point of Landsat trend in the same decade, between 1996 and 2005. According to a Portuguese drought report covering the period 1975–2006 [62], when our time series began, 1984–1989, the climate was particularly wet and was followed by a dry episode of 4 years (1990–1994). A more severe episode of drought occurred in 2002–2006. These events could explain the existence of change points in the same decade in study site trends, since cork and holm oak caliper development and photosynthesis are directly correlated to precipitation [18,63].

The trends also matched field knowledge and observations at a larger scale. A model of cork oak crown diameter applied to the Portuguese national forest inventory data from cork oak stands [17] determined a 9% decrease of cork oaks stands with a dense canopy cover (20 to 40%) between 1996 and 2006, in favor of sparse stands (less than 20% of cover). In our study, 12% of oak stands presented a significant decreasing trend.

Cork oak trends were estimated in southern Portugal for a 20 km$^2$ area in Serra de Grândola [27]. Landsat-derived EVI time series between 2000 and 2013 were extracted for five land cover types: dense cork oak stands with and without understory, sparse cork oak stands with and without understory, and grasslands. A Mann-Kendall test was used to determine significant trends of maximum EVI (in spring, from February to May) and minimum EVI (in summer, from July to September). Trends were hardly significant due to the short duration of the series, but the MK's Tau of each land cover type

revealed to be positive (increasing trend) for summer EVI trends, which is consistent with the main NDVI trends found in our study for the Grândola study site (Figure 5A,B). They also support the only-summer-NDVI methodology applied in our work by concluding that, while the upward trend detected in summer seemed to be driven by cork oak canopy, the decreasing trend found for spring was a consequence of shrub signal, even using an index less sensitive to the background like EVI.

Significant increasing NDVI trends (28.36% of total area) were expected in most of the stands, since the tree canopy extends and gets thicker each year. In the case of old stands, natural or artificial regenerations are supposed to efficiently replace dying trees with new ones, equilibrating NDVI values. On the other hand, significantly decreasing NDVI trends (11.78% of total area) can be a consequence of the drier and drier summers observed and predicted in southern Portugal [64]: one of the first tree responses to hydric stress is a decrease in photosynthesis activity [63], causing a progressive decline in NDVI values. Downward NDVI trends can also be due to the death of main branches or of the trees themselves. Besides the more severe climatic conditions such as droughts, some pests and diseases can also cause the decline and death of evergreen oaks, as a consequence of climate change, as the exhaustive review made by De Sampaio e Paiva Camilo-Alves et al. [65] shows. They analyzed the link between oak decline and the oomycete *Phytophthora cinnamomi*, an oak root pathogen that causes the same symptoms as drought and can lead to the death of the tree. The authors also pointed out the importance of other factors associated with oak decline, first of which was soil depth, compaction, and texture, which are directly impacted by cultural practices. Thus, oaks decline seems a complex multifactor process. By detecting summer stress and potential deaths, NDVI allows following efficiently the health and global yield of the stand.

The five land cover classes presented different proportions of significant decreasing NDVI trends. Costa et al. [66] described through three field examples the land abandonment and fragmentation of Portugal evergreen oak plots, due to the main changes in agricultural economic priorities and labor force availability since 1970, starting in the less productive stands (sparse holm oak stands). Then, the highest percentage of downward significant trends was found for cork oak agroforestry systems and the lower for holm oak agroforestry systems. In those systems, tree density is lower than in a forest structure and the canopy cover does not allow effective water retention in summer [6,67]. A comparison of co-occurring cork and holm oaks' response to summer drought showed that holm oaks have better adaptability to xerophytic climatic conditions, in accordance with their Portugal geographical distribution (Figure 2), possessing a more effective root system and being less vulnerable to embolism [11]; even though a study comparing tree-rings growth of *Q. ilex* in hotter and cooler Spanish sites suggests the drought adaption of this species has its limits and trees suffer from heat in the hotter sites [68]. Therefore, in the context of hotter and drier summers [64], physiological differences between the two species could have a long-term damaging effect on cork oaks and may benefit for now to holm oaks, especially in a low-density agroforestry system with low competitivity for water resources compared to a forest.

The Union of the Mediterranean Forest (UNAC, Portugal) classification of oak forests in Portugal based on soil and orography [69] allowed us to compare our trends with soil characteristics (Figure 8). The 'Tejo and Sado' river basins class is situated in arenite soils, with a more or less deep argic horizon, with reduced hydric retention and low fertility, with an altitude lower than 400 m. The 'Alentejo' type represents the large plains of the region, occupied by metamorphic or eruptive soils, more fertile and deep, with gentle slopes and an altitude level lower than 200 m. The 'Serra' (mountain in Portuguese) type has schist soils, an altitude higher than 200 m, and can present steep slopes. Figure 8 reveals that most declining stands are situated on the 'Serra'. This can be due to the soil quality [65] accentuated by severe climate episodes [62]. The model developed by the UNAC, however, points to another cause: a lower rentability of the 'Serra' type, which has more production cost for lower productivity. This decline would thus be a consequence of low levels of management and investment.

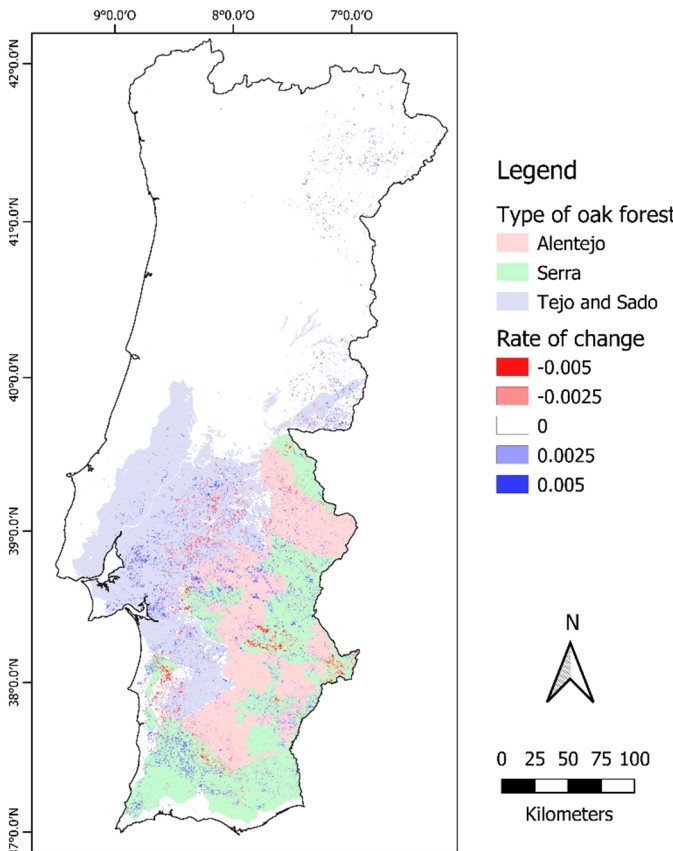

**Figure 8.** Comparison of the location of three types of oak forest and the TS rate of change (NDVIunit.year$^{-1}$) found for all CMK significant trends.

## 5. Conclusions

A methodology was successfully applied to map 34-year trends of NDVI of cork and holm oak stands at a spatial resolution of 30 meters through a whole country. Twelve percent of the total area was found to be declining (thirty percent of the area with significant trends). The use of six study sites permitted the validation of the results, as well as the exploration of the relationship between trends and productivity, and to find change-points in trends around the end of the 20th century. This work can be reproduced, improved, and adapted to other indices, areas, and vegetal species. The resulting trend map is the first step to identify the reasons of oak decline and anticipate the consequences of the inevitable climate change to come.

This work highlights the reliability of Landsat time series and the advantage for the scientific community to have this imagery for free. It also reveals the convenience of GEE to study long time series: a fast and easy acquisition of images and pre-processing (cloud mask), data extraction (calculation of vegetation indices), time series plotting, and pixel value extraction. Nevertheless, the platform remains limited to run specific tests (CMK and TS), to extract large images, or print heavy graphics. The procedure developed in GEE for a region and a specific type of forest, in our case Portugal and the cork and holm oak forests, can be easily adapted and updated for other regions and forest types and be used as a monitoring system of long-term changes in forest ecosystems.

**Author Contributions:** This investigation was realized by V.A., conceptualized and supervised by J.M.N.S. and J.A.P. Data imagery were collected, processed, and analyzed by V.A. Manuscript preparation was conducted by V.A. with inputs on analysis and content review and editing by J.M.N.S. and J.A.P.

**Funding:** V. Aubard was funded by the European Union Erasmus+ program (Erasmus+ Stages 2018, Vetagro-Sup, France) and the International student mobility grant of the Auvergne-Rhône-Alpes region, France. J.A. Paulo and J.M.N.Silva were funded by postdoctoral grants (SFRH/BPD/96475/2013 and SFRH/BPD/109535/2015,

respectively) from the Fundação para a Ciência e Tecnologia (FCT), Ministério da Ciência, Tecnologia e Ensino Superior, Portugal. The Forest Research Centre is a research unit funded by FCT (UID/AGR/00239/2013).

**Acknowledgments:** Authors acknowledge Companhia das Lezírias, Herdade da Contenda, and Instituto da Conservação da Natureza e das Florestas, for the access to study sites and for the use of facilities. They also thank Rita Andrews for her final English language editing.

**Conflicts of Interest:** The authors declare no conflict of interest. The funders had no role in the design of the study; in the collection, analyses, or interpretation of data; in the writing of the manuscript, or in the decision to publish the results.

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
