# Peer review of "Long-Term Monitoring of Cork and Holm Oak Stands Productivity in Portugal with Landsat Imagery"

_remotesensing, doi:10.3390/rs11050525_

Round 1

Reviewer 1 Report

The paper ‘Long-term monitoring of cork and holm oak stands productivity in Portugal with Landsat imagery’ deals with long-term trends analysis of the Normalized Difference Vegetation Index (NDVI) obtained from a time-series of Landast TM 5 and 7 ETM+ imagery in Portugal. Results were validated through MODIS NDVI time-series and by means of data derived from six representative cork oak stands.

I read with very interest the paper and consider promising the proposed methods that can be applied in different geographical contexts. The structure of the submitted manuscript is satisfactory in its present form while I consider proper the adopted methodology.

Prior to its acceptance, in the following rows I highlighted the minor revisions that the Authors should be provided to the manuscript.

About the introduction sections, what I suggest is to add something explaining the importance of cork oak woodlands also as significant forest ecosystem. While, referring to the conclusion section, I suggest also to highlight the reliability of Landsat time-series and the advantage for the scientific community to have these imagery for free.

Line 50-52 – this sentence need a reference as well as a reference year.

Lines 60-62 – the sentence ‘the normalized…’ should be referred with some bibliographic sources. In this direction, to have a comprehensive view, I would suggest to consider the book ‘The Normalized Difference Vegetation Index’ published in 2013.

Line 78 – Actually, the current 30 m of spatial resolution started with Landsat 4. Please change accordingly this sentence.

Lines 88-89 – In the sentence ‘The conclusion was 500 m pixels give …..’ something is missed. Please amend accordingly.

Line 152 – Add ‘data’ after ‘No COS’.

Line 189 – I suggest to add geographical locations (AML, Centro, etc.) also in the image on the right.

Line 204 – after its first use, the NDVI acronym must be used in the text.

Line 225-226 – Also in this case use the acronym GEE instead of Google Earth Engine.

Line 236 – in table 2, change ‘spatial scale’ with ‘spatial resolution’.

Lines 506-508 – ‘besides the more…’. I think that a bibliographic reference should be provided to this sentence.

Line 549 – ‘at a resolution of 30-m-squared’ must be changed as follows ‘at a geometrical resolution of 30 m’ or, if you prefer as follows ‘at a geometrical resolution of 30 m x 30 m’.

Author Response

Reviewer #1: The paper ‘Long-term monitoring of cork and holm oak stands productivity in Portugal with Landsat imagery’ deals with long-term trends analysis of the Normalized Difference Vegetation Index (NDVI) obtained from a time-series of Landast TM 5 and 7 ETM+ imagery in Portugal. Results were validated through MODIS NDVI time-series and by means of data derived from six representative cork oak stands.

I read with very interest the paper and consider promising the proposed methods that can be applied in different geographical contexts. The structure of the submitted manuscript is satisfactory in its present form while I consider proper the adopted methodology.

Prior to its acceptance, in the following rows I highlighted the minor revisions that the Authors should be provided to the manuscript.

About the introduction sections, what I suggest is to add something explaining the importance of cork oak woodlands also as significant forest ecosystem. While, referring to the conclusion section, I suggest also to highlight the reliability of Landsat time-series and the advantage for the scientific community to have these imagery for free.

Authors: We agree and improved Introduction and Conclusion sections accordingly.

Line 50-52 – this sentence need a reference as well as a reference year.

Authors: References were added.

Lines 60-62 – the sentence ‘the normalized…’ should be referred with some bibliographic sources. In this direction, to have a comprehensive view, I would suggest to consider the book ‘The Normalized Difference Vegetation Index’ published in 2013.

Authors: The suggested reference is relevant for this study and was included.

Line 78 – Actually, the current 30 m of spatial resolution started with Landsat 4. Please change accordingly this sentence.

Lines 88-89 – In the sentence ‘The conclusion was 500 m pixels give …..’ something is missed. Please amend accordingly.

Line 152 – Add ‘data’ after ‘No COS’.

Authors: Sentences were edited.

Line 189 – I suggest to add geographical locations (AML, Centro, etc.) also in the image on the right.

Authors: Geographical locations were added to Figure 2.

Line 204 – after its first use, the NDVI acronym must be used in the text.

Line 225-226 – Also in this case use the acronym GEE instead of Google Earth Engine.

Authors: Acronyms were used in the whole manuscript after their first use.

Line 236 – in table 2, change ‘spatial scale’ with ‘spatial resolution’.

Authors: Done.

Lines 506-508 – ‘besides the more…’. I think that a bibliographic reference should be provided to this sentence.

Authors: The exhaustive review made by De Sampaio e Paiva Camilo-Alves et al. was considered an adequate bibliographic reference.

Line 549 – ‘at a resolution of 30-m-squared’ must be changed as follows ‘at a geometrical resolution of 30 m’ or, if you prefer as follows ‘at a geometrical resolution of 30 m x 30 m’.

Authors: We chose to change for ‘at a spatial resolution of 30 m’.

Reviewer 2 Report

This manuscript presents an interesting and effective methodology for long-term monitoring of productivity in cork and holm oak stands.  The methods employed by the authors are sound and comprehensive and this paper will provide a valuable contribution to the literature.  The paper could benefit from a final English language editing effort to correct the remaining grammatical inconsistencies.  I added some suggestions for the authors in the form of comments on the original manuscript.  In a few cases these suggestions may not conform to the original intent of the authors, but they will at least point out an area where clarification should be beneficial.  The authors should not assume that my suggestions are comprehensive in this regard, but rather use the suggestions wherever they might apply, as they (and perhaps an English editor) go through the entire manuscript one last time. 

Author Response

Reviewer #2: This manuscript presents an interesting and effective methodology for long-term monitoring of productivity in cork and holm oak stands.  The methods employed by the authors are sound and comprehensive and this paper will provide a valuable contribution to the literature.  The paper could benefit from a final English language editing effort to correct the remaining grammatical inconsistencies. I added some suggestions for the authors in the form of comments on the original manuscript.  In a few cases these suggestions may not conform to the original intent of the authors, but they will at least point out an area where clarification should be beneficial.  The authors should not assume that my suggestions are comprehensive in this regard, but rather use the suggestions wherever they might apply, as they (and perhaps an English editor) go through the entire manuscript one last time. 

Authors: The corrections suggested in the original manuscript were accepted and adapted, if necessary, to keep and clarify the original intent. A final American English language editing was performed by a native speaker.